# A Fully Integrated Bluetooth Low-Energy Transceiver with Integrated Single Pole Double Throw and Power Management Unit for IoT Sensors

**DOI:** 10.3390/s19102420

**Published:** 2019-05-27

**Authors:** Sung Jin Kim, Dong Gyu Kim, Seong Jin Oh, Dong Soo Lee, Young Gun Pu, Keum Cheol Hwang, Youngoo Yang, Kang Yoon Lee

**Affiliations:** Department of Electrical and Computer Engineering, Sungkyunkwan University, Suwon 16419, Korea; sun107ksj@skku.edu (S.J.K.); rlarlarbrb@skku.edu (D.G.K.); geniejazz@skku.edu (S.J.O.); blacklds@skku.edu (D.S.L.); hara1015@skku.edu (Y.G.P.); khwang@skku.edu (K.C.H.); yang09@skku.edu (Y.Y.)

**Keywords:** bluetooth low-energy (BLE), gaussian frequency-shift keying (GFSK), transceiver (TRX), low power, one-point modulation, automatic bandwidth calibration (ABC), single pole double throw (SPDT), DC-DC buck converter

## Abstract

This paper presents a low power Gaussian Frequency-Shift Keying (GFSK) transceiver (TRX) with high efficiency power management unit and integrated Single-Pole Double-Throw switch for Bluetooth low energy application. Receiver (RX) is implemented with the RF front-end with an inductor-less low-noise transconductance amplifier and 25% duty-cycle current-driven passive mixers, and low-IF baseband analog with a complex Band Pass Filter(BPF). A transmitter (TX) employs an analog phase-locked loop (PLL) with one-point GFSK modulation and class-D digital Power Amplifier (PA) to reduce current consumption. In the analog PLL, low power Voltage Controlled Oscillator (VCO) is designed and the automatic bandwidth calibration is proposed to optimize bandwidth, settling time, and phase noise by adjusting the charge pump current, VCO gain, and resistor and capacitor values of the loop filter. The Analog Digital Converter (ADC) adopts straightforward architecture to reduce current consumption. The DC-DC buck converter operates by automatically selecting an optimum mode among triple modes, Pulse Width Modulation (PWM), Pulse Frequency Modulation (PFM), and retention, depending on load current. The TRX is implemented using 1P6M 55-nm Complementary Metal–Oxide–Semiconductor (CMOS) technology and the die area is 1.79 mm^2^. TRX consumes 5 mW on RX and 6 mW on the TX when PA is 0-dBm. Measured sensitivity of RX is −95 dBm at 2.44 GHz. Efficiency of the DC-DC buck converter is over 89% when the load current is higher than 2.5 mA in the PWM mode. Quiescent current consumption is 400 nA from a supply voltage of 3 V in the retention mode.

## 1. Introduction

Recently, the Internet of Things (IoT) can be applied to various applications such as wearable devices, sensor networks, and health care [1]. One of the essential requirements of the IoT application is the low power wireless connectivity for the long battery life. The bluetooth low-energy (BLE) standard is a promising wireless connectivity for IoT applications [2]. The BLE operates in the 2.4-GHz industrial, scientific, and medical (ISM) band and uses GFSK with a modulation index *h* = 0.5 for signaling. Nominal frequency deviation, *f_dev_*, shall be derived as Equation (1) [2].
(1)f_dev (kHz)=((data_rate×h))/2
Therefore, *f_dev_* is ± 250 kHz when the data rate is 1 Mbps.

Reference [3,4] proposed a method that can support a high data rate. Reference [3] proposed a PLL using 2-point modulation, and Reference [4] used a direct-conversion transmitter structure. Even though the 2-point modulation scheme or direct-conversion architecture can achieve high data rates, compared with the 1-point modulation, the additional digital analog converter (DAC) circuits and up-conversion mixers will increase current consumption and area. This paper proposes a method with wide bandwidth using one-point modulation for a low power operation. The Automatic Bandwidth Calibration (ABC) method has stable PLL bandwidth for the frequency channel, process, voltage, and temperature variation. In order to improve receiver sensitivity and the linearity of the receiver, the Auto Gain Control (AGC) function is essential. The preamble time of the BLE receiver standard is 8us [2]. Therefore, the receiver must finish a whole gain setting within this time through AGC. In Reference [5], a 2-step Fast AGC method is proposed to reduce the AGC time. However, 2-step Fast AGC requires a high-resolution ADC, which will increase current consumption and area. Gain settling time of RX is most affected by the group delay of Base Band Analog (BBA). In this paper, we propose a method to minimize the filter group delay by changing the bandwidth of the DC offset cancellers (DCOC) automatically during gain change. Generally, if the duty of the Local oscillator (LO) signal is designed to be 50%, two Low Noise Amplifiers (LNAs) should be used because of the noise characteristic of the RF front-end (RF-FE) degrades at the overlap of the LO signal [6]. However, this paper proposes a high-gain, low-noise, low-power RF-FE that can only use an inductor-less low noise transconductance amplifier (LNTA) structure with LO of 25% duty ratio. Because the BBA operates at low power, the characteristics of the filter are sensitive to process, voltage, and temperature variations [7]. In this paper, we propose a Filter Tuning Circuit (FTC) to keep the cut-off frequency and bandwidth of the filter constant. The ADC is designed with a fully differential 6-bit successive approximation register (SAR) ADC architecture. To reduce current consumption, Adaptive Power Control (APC) is applied to the dynamic latched comparator. External components such as inductor and capacitor for impedance matching, baluns, and a transmitter (TX) as well as a receiver (RX) switch can increase system cost and be area consuming on the PCB. Therefore, it is beneficial to design the BLE RF transceiver that can interface directly with a single-ended 50-Ω antenna minimizing external components [7]. In this work, Single-Pole Double-Throw (SPDT), internal matching components, and Power Management Unit (PMU) such as the DC-DC buck converter and Low Drop-Outs (LDO) are integrated to reduce system cost. This paper presents the low power, low-cost RF transceiver that can achieve excellent receiver sensitivity, dynamic range, and transmitter spurious performance.

This paper is organized as follows. Section 2 describes the proposed RF transceiver architecture for the BLE application. Section 3 presents building blocks of a low power FSK transceiver. The experimental measurements conducted on a test chip from the 1P6M 55 nm CMOS implementation are described in Section 4. The conclusion is presented in Section 5.

## 2. BLE Transceiver Architecture

Figure 1 shows a block diagram of the proposed transceiver for the BLE application. It is composed of a frequency synthesizer using an analog PLL, the transmitter with class-D RF PA, an integrated SPDT switch, and the PMU with DC-DC buck converter in triple-mode and the LDOs. Since the on-chip SPDT switch eliminates the need for external RF components between chip and antenna, it is possible to configure the minimum-sized modules. It is critical that RF application of BLE require PCB area to be as small as possible. Therefore, sharing matching components of TX and RX front-end can be beneficial in terms of area and cost. Architectures of TX and RX are analog PLL-based on direct modulation instead of IQ up-conversion, and low IF down-conversion, respectively. Although the digital PLL can have the benefit in terms of the area, analog PLL is implemented in this paper to avoid the complexity of additional calibration logics in digital PLL [8]. The proposed TX uses the inductor-less class D type PA for small area and high efficiency [9]. It requires bandwidth of PLL of 1 MHz due to PLL based on direct modulation of 1 Mbps [10]. The proposed low-IF RX uses an inductor-less LNTA, a passive quadrature down-conversion mixer, and the trans-impedance amplifiers (TIA).

## 3. Building Blocks

### 3.1. RX Front-End

Figure 2 shows the schematic of RX RF-FE. The RF-EF is composed of an LNTA, passive mixers with the 25% duty generator, and TIAs. In general, when using a 50% duty cycle LO signal, two LNTAs should be used because the LO signal is overlapped and deterioration of the noise feature of the RF-FE [6]. However, to reduce the power consumption, only a single-ended LNTA with 25% duty-cycle LO is used instead of two LNTAs in this paper. A 25% duty cycle LO increases 3-dB conversion gain, which lowers the noise contribution of the mixer compared to 50% duty-cycle LO. Since there is no overlapped period between LOs, an LNTA can drive IQ passive mixers without performance degradation. Additionally, current mode operation is adopted for high gain and low noise. The proposed inductor-less LNTA is a combination of a self-biased inverter type amplifier and separated biasing (V_B1_). This architecture is more suitable for low power structure than only self-biasing technique because it has a high gain even though the sizes of MOSFETs are small [11].

Noise figure and gain matching of LNTA are well optimized by controlling V_B1_. Impedance matching of RX is realized by sharing SPDT with TX. Although SPDT provides some isolations, RX should consider impedance of TX when turned off. The well estimated TX impedance can mitigate degradation of performance in RF-FE. It guarantees RX to have a sufficient gain and low noise figure at the desired RX band. In a BLE standard, the linearity of mixers is a significant issue because of the process of interferers [2]. Passive mixers are used in this work with a low supply voltage operation. A current conveyer structure is used for the LNTA and passive mixer. C_M_ prevents the image current when the switches are turned on simultaneously by overlap between 25% duty cycle LO signals.

### 3.2. Phase-Locked Loop (PLL) for RX LO Generator and TX Modulator

Figure 3 shows the block diagram of proposed analog PLL. It is composed of VCO, dividers (/2,/2-/3 and Fractional-N Divider with SDM), PFD, CP, internal third order loop filter, ABC, and the GFSK modulator.

Bandwidth requirement of the PLL is that is wider than the data rate for PLL based on one-point modulation in the TX mode [10]. Therefore, bandwidth is used about 1 MHz or more. However, bandwidth of approximately 200 kHz was used to meet the specification of bandwidth of the BPF (1 MHz ± 600 kHz) in an RX mode.

The bandwidth of PLL with a 3rd order loop filter can be written as Equation (2) [3].
(2)$$\omega_{C}=\left(\frac{K_{VCO}\cdot I_{CP}\cdot R_{1}}{2\pi N}\right)\times \left(\left(\left(R_{1}+\frac{1}{C_{1}}\right)||\frac{1}{C_{2}}\right)||\left(R_{2}+\frac{1}{C_{3}}\right)\right)$$
where *ω_C_* is the bandwidth of the PLL, *K_VCO_*, *I_CP_*, and *N* are the gain of VCO, and the current of CP, and division ratio of the divider, respectively. Additionally, *R_1_*, *R_2_*, *C_1_*, *C_2_*, and *C_3_* are resistors and capacitors of the third order loop filter. In Equation (2), the bandwidth is determined from parameters of the third order loop filter.

Table 1 shows the PLL loop parameter according to the loop bandwidth depending on the TX and RX mode. After determining parameters of the loop filter, the target frequency and bandwidth of PLL are determined with ABC. To adjust the desired bandwidth, the frequency of VCO is calibrated to target frequency and measure the *K_VCO_*. The *I_CP_* can be calculated by applying the measured *K_VCO_* and fixed *N* value to Equation (2). As can be seen from the table, the TX and RX bandwidths are different from each other. The PLL lock time is proportional to the 4/bandwidth, which makes the RX PLL locking time about 20 μs [12]. Since the TX/RX switching time is 150 µs from the BLE specification [2], the RX PLL locking time can satisfy the BLE specification with the certain margin.

Figure 4a,b show a block diagram and timing diagram of the ABC block, respectively. The ABC block is composed of a 12-bit Counter, a Finite-State Machine (FSM), a Digital Comparator, a VCO Frequency Tuning Controller, and BW Calculator. ABC block works with the reference clock signal (CLK_REF_) to generate a Reset Counter (CNT_RST_), Enable Counter (CNT_MASK_), Decision Counter (CLK_TUNE_), and Comparison Clock (CLK_COMP_) signals through FSM. Since the frequency tuning process is frequency tracking, a digital accumulator is used to estimate the period of the VCO.

The ABC block is operated by two-step as follows.

Step 1: metal-oxide-metal (MOM) capacitances of the cap bank of VCO are controlled by VCO_CAP_<9:0> in this step. Optimum MOM capacitances are selected through a VCO Frequency Tuning Controller. Free-running frequency of the VCO is near the target channel frequency after the frequency tuning of VCO is completed.

Step 2: Calculation of the bandwidth begins when frequency tuning of the VCO is completed. The K_VCO_ is defined as the frequency range of the VCO with respect to a V_CTRL_ change. Thus, V_CTRL_ is changed by changing VC<1:0>. K_VCO_ is calculated as Equation (3).
(3)$$K_{VCO}=\left(\frac{CNT_{VAL.VC=11}-CNT_{VAL.VC=01}}{V_{CTRL}}\right)\times 4$$

When TX modulation is enabled and frequency is changed abruptly, the spurious emission mask can’t be met at output of the transmitter due to harmonic tones. When input of TX Data is ‘0’ or ‘1’, the modulation deviation value is added or subtracted from carrier frequency. By mapping and filtering, levels of spurious tones can be reduced by changing inputs to the DSM in PLL. 

The Fractional-N Divider is composed of the pulse-shallow counter with divide-by-4/5 and 3-order DSM. Output frequency of the proposed analog PLL is calculated as Equation (4).
(4)$$Frequency=PRE_{DIV}\times \left(4PC+SC+\frac{MC\pm F_{DEV}}{2^{22}}\right)\times CLK_{REF}$$

The value of *PRE_DIV_* is 3 and 2 when the channel frequency is channel-15 and other channels, respectively. Values of *PC*, *SC*, and *MC* are determined in the channel table of PLL with respect to the channel value (*CH*). The value of *F_DEV_* is 10,923 and 16,384 when channel frequency is 15-channel and other channels, respectively. The modulation value adds or subtracts the value of *F_DEV_* to the value of *MC* when *TX_DATA_* is ‘1’ and ‘0,’ respectively.

Figure 5 shows proposed VCO with MOM capacitor bank. The proposed VCO is designed with MOM capacitors in this paper. They are stacked from Metal 3 to Metal 5 for high capacitor density and reduced die area [13].

### 3.3. Low IF Base-Band Analog

The low-IF RX requires a block for image rejection. Proposed receiver structure uses the pair of the complex BPF for rejecting the image band. Figure 6 shows the block diagram of the proposed low-IF BBA. It is composed of three stage VGAs, two 2nd order BPFs, and three DCOC [14]. Total dynamic range of BBA is 88 dB. The gain of VGAs (VGA1, VGA2, VGA3) and BPFs (BPF1, BPF2) are 20 dB and 14 dB, respectively. A high pass filter (HPF) is used for the DCOC. According to the amplitude of RF signal changes the gain of BBA.

The baseband produces a constant output voltage for the ADC input range during AGC timing. The proposed BLE receiver uses VGA to achieve constant baseband output. The gain of VGAs is adjusted by using the resistor ratio. An AGC is proposed to control gain of the BBA automatically during the preamble duration of eight symbols (8 µs) [2]. Preamble duration is too short, which is 8 μs in BLE specification. Settling-timing of AGC is determined by group delay of BBA. Bandwidth of DCOC (BW_DCOC_) is dominant in group delay of BBA. If BW_DCOC_ is too wide at the Measure_AGC period to reduce group delay, MODEM achieves invalid information gain since output of BBA is too attenuated. Conversely, if BW_DCOC_ is too narrow to achieve characteristics of attenuation, MODEM achieves invalid information of gain since output of BBA cannot be settled down properly. Therefore, the BW_DCOC_ can be controlled by the DCOC controller so that bandwidth becomes wide enough during preamble duration to operate the AGC loop in this paper.

Figure 7 shows the timing diagram of the proposed DCOC controller according to operation of AGC. The AGC should control the gain of BBA within 2 µs by the MODEM. Therefore, gain of BBA can be controlled a maximum of three times since the preamble time is 8 μs.

First, during the ED interval in AGC_G1, input level of −50 dBm is detected by the peak detector, and the initial gain is set by MODEM. In AGC_G2, the AGC starts and the gain value of BBA is determined coarsely. If the BBA output of the level cannot reach the desired level, the gain value of BBA is determined finely in AGC_G3. After AGC_G3, the gain value is set.

In the AGC operation, the BW_DCOC_ should be changed to guarantee completion of AGC within 8 µs by controlling the R_DCOC_<1:0> [2]. The BW_DCOC_ is 2.5 MHz during 36 clocks of CK, and BB_OUT_ is settled fast since the gain information of BBA is not critical at Change_AGC period. After 36 clocks, the BW_DCOC_ is changed to 350 kHz since the 3-dB bandwidth of BBA is from 400 kHz to 1.6 MHz. When the AGC operation is finished, the freeze signal of AGC (AGC_FR_) becomes high. The BW_DCOC_ is changed to 100 kHz that does not affect the bandwidth of BBA. As shown in Figure 7, if the BW_DCOC_ is fixed, BBA gain cannot be determined accurately since the common mode of outputs of the BBA is not settled properly. If the settling is not completed during the preamble period, data errors can occur.

As shown in Figure 8, the pseudo differential structure is used for the VGAs for infinite input impedance. Gain steps of VGA1, VGA2, and VGA3 are 4 dB, 2 dB, and 1 dB, respectively [15]. The VGAs use a differential to a single two-stage amplifier. Because input impedance of VGA is high, it does not affect the previous stage. The VGA is designed to have wide dynamic range and its gain is controllable by the modem system. Gain control scheme using the resistor bank is used in this design. Gain of VGA is controlled digitally by a digital modem.

Gain range is from 0 dB to 60 dB. The gain (A_V_) of VGA is determined by the ratio of R_1_ and R_var_, as shown in Equation (5). Since the gain is controlled by the relative ratio of resistors, the error of the gain is small depending on PVT variations.
(5)$$A_{V}=1+\frac{R_{1}}{R_{var}}$$

Figure 9 shows the designed 2nd-order Chebyshev complex BPF. To achieve the complex operation, it uses in-phase signal and quadrature signal. Characteristic of BPF is made by the low pass filter (LPF) characteristic shifting DC to IF using a cross coupled resistor [14]. To reduce process variation, capacitor arrays are composed of capacitor and MOS switches. It controls the bandwidth of the BPF and its control signals (C_BPF_) are determined by FTC. Center frequency of the complex BPF is 1 MHz and 3-dB bandwidth is 1.2 MHz. Image frequency rejection ratios is 36 dB [15].

Figure 10 shows a schematic of the FTC. It must compensate a capacitance and resistance variation according to process variation. It is composed of current mirror and capacitor array for generating charging voltage (V_CH_) and comparator and filter tuning controller. If values of resistors and capacitors are changed by process variation, the charge time of V_CH_ is changed. After then, filter tuning controller compares V_CH_ charge time with reference charge time. In this paper, the capacitor is only tuned because resistor variation is reflected in I_REF_ variation. Resistance of R_REF_ is the same as bandwidth resistor value of BPF to apply resistance variation of the same ratio. It can reduce the tuning time and die area.

Figure 11 shows the timing diagram of FTC. When CH_ON_ signal is high, the V_CH_ is increased by charging the BPF CAP BANK replica. Output of the Comparator (OUT_COMP_) becomes high when V_CH_ is higher than V_REF_. If the OUT_COMP_ is low when COMP_CLK_ is high, the value of C_BPF_ is decreased by FTC and start the calibration loop again. Therefore, if the OUT_COMP_ is high when COMP_CLK_ become high, the value of C_BPF_ is increased. In addition, if the OUT_COMP_ value is changed by comparing the value of the previous state, the calibration is finished to reduce the tuning time. After calibration, the determined value of C_BPF_ is applied to the capacitor array of two 2nd order complex BPFs. The maximum tuning time is 150 μs of 32 cycles. If the tuning process is completed, the FTC is turned off to save power consumption.

### 3.4. Analog to a Digital Converter

Figure 12 proposes the designed 6-bit fully differential SAR ADC structure. The resolution of ADC required in modem requires 5-bit, but was designed with 1-bit margin when designing the SAR ADC. Input signals of the ADC include V_I_, V_IB_, and V_Q_, as well as V_QB_ that are differential inputs. Therefore, two parallel ADCs should be applied in the proposed receiver. Each ADC is composed of a comparator and two binary-weighted capacitor arrays. The SAR logic controls the switching sequence of these ADCs. The fully differential structure of the ADCs reduces the substrate and supply voltage noise, and has a good Common Mode Rejection Ratio (CMRR) [16].

Capacitor arrays of this ADC operate as sample and hold circuits and DACs. Significant power consumption of the SAR ADC may occur due to switching in the capacitor array. The switching sequence of the proposed structure is common mode voltage (V_CM_)-based and straightforward. Previous works have proven the V_CM_-based straightforward ADCs as one of the most energy efficient structures [17].

In the V_CM_-based switching and after sampling, in each cycle, one of the capacitors of the capacitor array switches from V_CM_ to the Reference Voltage (V_REF_) or 0, according to the comparator decision. This switching voltage value is half of the one in the conventional SAR ADC structures. Therefore, switching power consumption is reduced significantly. Switching is straightforward, which means that only the next capacitors will be switched, and previously switched capacitors will not switch until the current switching cycle finishes and the next switching cycle starts. This sequence of switching minimizes switching steps and reduces power consumption. As voltages across capacitors are changed only from V_CM_ to V_REF_ or from V_CM_ to 0 V, charging and discharging the time of capacitors decreases, which is useful when conversion speed of the ADC increases.

Figure 13 shows the dynamic latched comparator applied in the ADC. It is composed of the pre-amplifier and dynamic latch to prevent kick noise. The pre-Amplifier has N-type and P-type differential input pair for rail-to-rail input range. Power efficiency of the conventional dynamic latched comparator is poor due to the static current even after comparison operation. In this work, through the APC logic after pre-amplifying and output decision, static current consumption of the pre-amplifier is blocked, which improves the power efficiency of the ADC [17].

### 3.5. TX Power Amplifier

Figure 14 shows the proposed PA with SPDT. It consists of the 16-PA Unit cell with Ramping Controller, and SPDT. The proposed PA is implemented as Class-D type instead of Class-E type considering breakdown voltage of the device and its efficiency. When the PA is enabled or disabled abruptly for data transmission, undesired harmonic tones can be generated.

Since the undesired harmonic tones can degrade spurious spectral emission characteristic of the TX [8], the proposed PA is divided to 16-PA Unit cells, and it applies the Ramping Controller to digitally control output power with ramping when PA is enabled or disabled. The SPDT is designed and integrated for single antenna to be connected to the TX path or the RX path. The body floating technique is adopted to reduce insertion and isolation loss [18].

### 3.6. DC-DC Buck Converter and LDO

Figure 15 shows a block diagram of the proposed triple-mode DC-DC buck converter. It is composed of a bandgap reference (BGR), Power MOSFETs, self-calibration negative current detector (SC-NCD), and the triple-mode (PWM, PFM, and retention) controller. Each mode has different characteristics to achieve the wide load current range. The PWM mode controller is designed to operate in the active state to provide good regulation characteristics of V_OUT_ with low output ripple. Retention and PFM mode controller are designed to reduce switching losses and internal current consumption since the operation is a sleep or stand-by state [19]. The PWM mode and PFM mode is operated under the load of over 2.5 mA and the load of between 0.5 mA and 2.5 mA, respectively. The retention mode is enabled to improve the efficiency when the low load current is below 0.5 mA.

The DC-DC buck converter generates the output voltage of 1.2 V from the supply voltage of 1.5 V to 3.6 V. The operation is defined by the connection of the external 1 μH inductor (L_1_) and the 1 μF capacitor (C_1_). To obtain high conversion efficiency in a discontinuous conduction mode (DCM), the SC-NCD adjusting the NMOS switch (M_N_) off-time is proposed. By controlling M_N_ off-time, both diode conduction losses in power MOSFET are minimized effectively. The 1.2 V output of the DC-DC Buck converter supplies a BGR and four on-chip LDOs composed of three low-noise LDO with a capacitor and a capacitor-less LDO, each making a 1 V core supply voltage for the different blocks of the transceiver [20].

Figure 16 shows the block diagram of the proposed SC-NCD. It is a fully digital controller composed of UP/DN counter, duty controller, and D-FF. It is smaller than current consumption of conventional NCD. The proposed SC-NCD method does not use the comparator when it detects the point that the inductor current is 0 A. Therefore, the SC-NCD can improve efficiency by reducing control loss account for a large proportion of the DCM dc-dc converter in light load current conditions using the NCD for digital methods, and it is possible to prevent efficiency reduction due to NCD timing errors caused by the analog comparator offset.

Figure 17a,b show timing diagram of operation of NCD. VX signal is sampled to digital bits by D-FF. The M_N_ is turned off after 3 ns and 12 ns. The sampled bits, S1 and S2, feeds to logic including XOR, NOR, and the AND gate and output signal of that is feed to UP/DN counter to control M_N_ off-time digitally. In the case where DRV_N_ is under-duty, sampled bits, S1 and S2, are all high and converting the UP signal zero to high. If the DRVN signal is over duty, sampled bits are all-zero and converting the DN signal zero to high. Lastly, the STAY signal is set to high and duty of CK_N_ is locked.

Figure 18 shows block diagram of low noise LDO with a fast settling technique (FST). The low noise LDO helps minimize the VCO phase noise and reduce the impact of VCO pushing [21]. It is composed of LPF, a BGR, an LDO, and an NMOS for FST. The LPF is used to reduce output noise of BGR. The FST is composed to compensate settling time, which slowed using LPF of BGR. When EN_LDO_ is low in an initial state, the FST is enabled. The M_F1_ and M_F2_ are turned on by SET_FAST_. Therefore, the R_LPF_ is shorted to achieve fast settling of V_REF_. The LDO output capacitor (C_LDO_) is initially charged by using the bypass mode of LDO during fast settling time.

Figure 19 shows timing diagram of FST. When the BGR is enabled at T_1_, the R_LPF_ is bypassed by turning on switching MOSFET (M_F_). Therefore, the reference voltage of the LDO is applied without delay and C_LDO_ is charged in the bypass mode. It can be rapidly settled. When the LDO is enabled at T_2_, output voltage of BGR is applied to the LDO through the LPF by turning off M_F_, and noise of LDO is reduced. Output voltage of the LDO is changed to 1 V from VDD (1.2 V).

## 4. Experimental Result

Figure 20 shows a micro-photo of the proposed transceiver for BLE application designed in 1P6M 55-nm CMOS process. The die sizes of transceiver are 1.44 × 1.14 mm^2^ and that of DC-DC buck converter is 0.45 × 0.33 mm^2^ including the ESD protection pads.

Figure 21 shows a measurement board of the proposed BLE transceiver. It is composed of a BLE transceiver IC, a 32-MHz crystal, a 50 Ω-antenna, TRX matching network, and an LC filter of the DC-DC converter.

Figure 22a,b show measured phase noise results of PLL in TX mode and RX mode, respectively. Bandwidth of PLL is 1 MHz and 200 kHz in the TX mode and RX mode, respectively. Frequency offset of TX and RX is 1 MHz at the same channel since proposed TRX is a low-IF structure.

The S11 characteristics of RX including SPDT switch is below −10 dB with respect to BLE channels, as shown in Figure 23.

Figure 24 shows the noise figure and IIP3 measurement results with respect to BLE channel frequencies. The noise figure and IIP3 is up to 6.52 dB and −17.8 dBm, respectively.

Figure 25 shows measured gain and bandwidth of BBA. The IF and 3-dB cut-off bandwidth of the BBA are 1 MHz and 1.2 MHz, respectively. The BBA has the high, mid, and low gain of 88 dB, 38 dB, and −8 dB, respectively. Rejection ratios at the image frequencies of −1 MHz and −2 MHz are 36 dB and 42 dB, respectively.

Figure 26 shows the transient simulation results according to the gain change of BBA with the DCOC Controller. The BW_DCOC_ is 2.5 MHz during 36 clocks of CK, and BB_OUT_ is settled fast. After 36 clocks, the BW_DCOC_ is changed to 350 kHz.

Figure 27a,b show measurement results of ENOB and INL/DNL of ADC, respectively. The ENOB and INL/DNL of ADC are 5.69-bit and ± 0.5 LSB, respectively.

Figure 28 shows measured output spectrum characteristics of Tx. Fundamental output power is approximately −0.39 dBm, and the second and third harmonic tones are −50.46 dBm and −47.47 dBm, respectively.

Figure 29a,b show the spurious emission mask of TX and demodulated TX Frequency for the BLE packet. The BLE specification requires spurious emissions from 1 to 1.5 MHz offsets at carrier frequencies to be at least 20 dB lower than the main power, and adjacent channel power for channels above 2 MHz from the carrier frequency should not exceed −20 dBm [2]. The TX output spectrum can satisfy the reference emission mask in the specification. The average measured modulation deviation corresponding to the alternating “10” data pattern is 207 kHz, which creates the most Inter Symbol Interference (ISI). It guarantees the margin of 22-kHz above the specification of 185-kHz, as shown in Figure 29b [2].

Figure 30a,b show measurement waveforms of the triple mode DC-DC converter depending on load current in the PWM/PFM and retention modes, respectively. Switching frequency and duty of the PFM mode is 1 MHz and 45 ns under the load current of 2 mA, respectively. When load current is over 2.5 mA, the operation mode of the DC-DC converter is automatically changed to the PWM mode. Switching frequency and duty of PWM mode is 2.5 MHz and 120 ns under the load current of 5 mA, respectively. When the load current of the DC-DC converter is below 0.25 mA, the operation mode of the DC-DC converter is automatically changed to the retention mode. Output is 1.2 V with the ripple of 20 mV in the retention mode. Output ripple has the frequency component of 32-Hz, which is the same as the switching frequency, and the switch turns on for the time of 1/1024 every duty cycle.

Figure 31 shows the efficiency of the DC-DC buck converter in the triple-mode [20]. The DC-DC buck converter can operate under the load current from 1 μA to 10 mA and has maximum efficiency of more than 89%. The mode is changed from the PFM mode to the PWM mode at 2.5 mA because the efficiency of the PWM mode is higher than that of the PFM mode at 2.5 mA.

Figure 32a,b shows simulated output noises of BGR and LDO, respectively. Output noises of BGR and LDO are 5.9 nV/Hz and 23 nV/Hz at the frequency of 100 kHz, respectively. 

Figure 33a,b show measured output of the LDO with and without FST, respectively. If the BGR and the LDO are turned on without FST, the settling time is 320 μs, as shown Figure 33a. Since the resistor of LPF (R_LPF_) is bypassed by turning on switching MOSFET (M_F_) in Figure 19, the reference voltage of the LDO is applied without delay, and the output capacitor of LDO is charged in bypass mode. It can be quickly settled. Therefore, the settling time of BGR and LDO with FST is 29 μs, which is 10 times faster than without it, as shown in Figure 33b.

Table 2 shows the performance comparison with recent BLE transceivers. This work is implemented in the 55-nm CMOS process including TRX switch and DC-DC buck converter. The die area and power consumption of this work is smaller than that of References [3,4]. The power consumption is smaller than those of other works achieving the similar Rx performance (noise figure and sensitivity) and TX output power even though the die area is larger than in Reference [21].

## 5. Conclusions

This paper presents a low power FSK TRX with an integrated SPDT switch and high efficiency power management unit for BLE application. It is implemented with the RF front-end with an inductor-less LNTA and 25% duty-cycle current-driven passive mixers, and low IF baseband analog with complex BPF to reduce power consumption and area, and improve the image rejection ratio, respectively. In the analog PLL, low power VCO is designed by using ABC. This is proposed to optimize bandwidth, settling time, and phase noise by adjusting the charge pump current, VCO gain, and resistor and capacitor values of the loop filter. Current consumption of the ADC is reduced by adopting straightforward architecture. The GFSK modulation is implemented to ensure the proposed low power transceiver can operate at the data rate of 1 Mbps. The DC-DC Buck converter improves overall efficiency by automatically selecting optimum mode among triple modes, PWM, PFM, and retention, depending on the load current. The low noise LDO is designed to improve receiver sensitivity and phase noise of VCO. The transceiver is implemented using 1P6M 55-nm CMOS technology and the die area is 1.79 mm^2^. Power consumption of the receiver and transmitter are 5 mW and 6 mW from the supply voltage of 3V, respectively. Noise figure of the receiver is up to 6.5 dB with respect to channel frequencies. Measured sensitivity of Rx is -95 dBm at 2.44 GHz. The measured phase noise of the PLL is −87.1 and −112.2 dBc/Hz at 100 kHz and 1 MHz offset from 2.44 GHz in the receiver mode, respectively. Efficiency of the DC-DC buck converter is over 89% when the load current is higher than 2.5 mA in the PWM mode. Quiescent current consumption of the TRx is 400 nA from a supply voltage of 3 V in the retention mode.

## Figures and Tables

**Figure 1 sensors-19-02420-f001:**
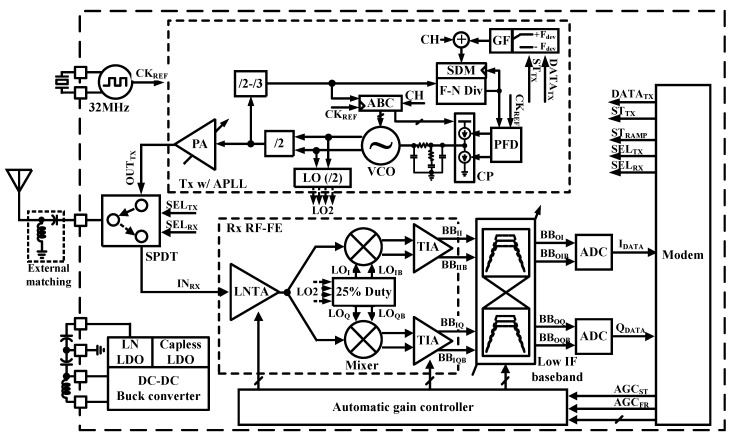
Block diagram of the proposed BLE transceiver with a Power Management Unit and Single Pole Double Throw.

**Figure 2 sensors-19-02420-f002:**
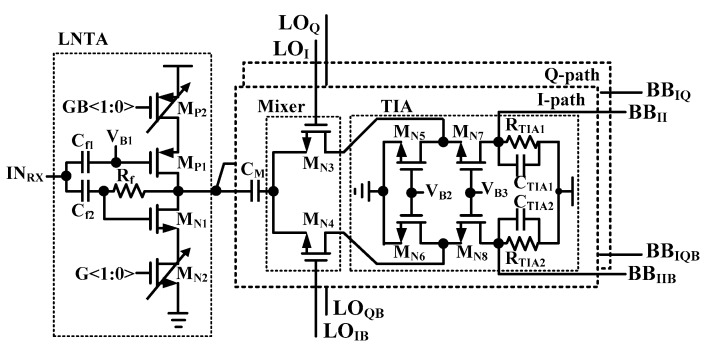
Schematic of RF-FE with 25% duty-cycle current-driven passive mixers.

**Figure 3 sensors-19-02420-f003:**
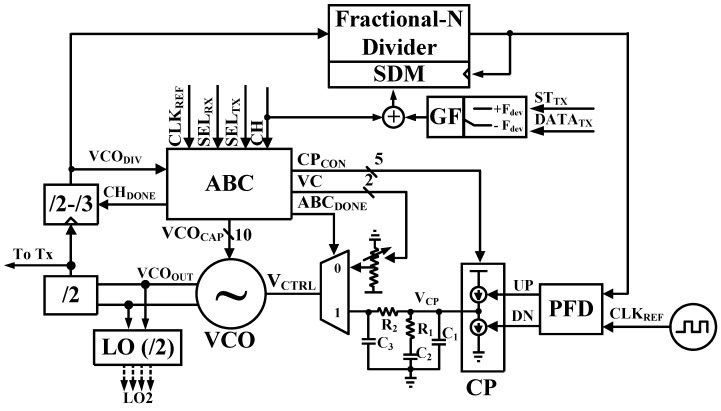
Block diagram of the analog phase-locked loop with automatic bandwidth calibration.

**Figure 4 sensors-19-02420-f004:**
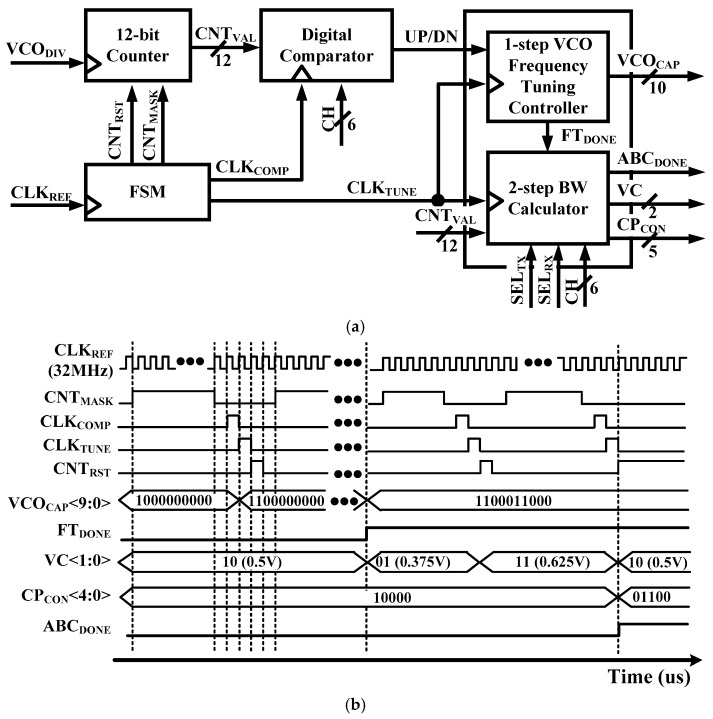
(**a**) Block diagram and (**b**) timing diagram of the ABC block.

**Figure 5 sensors-19-02420-f005:**
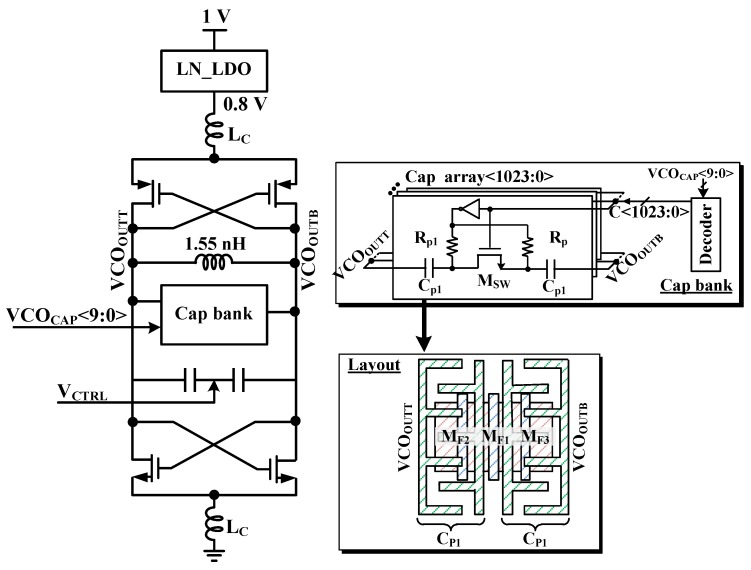
Schematic of VCO with MOM Capacitor bank.

**Figure 6 sensors-19-02420-f006:**
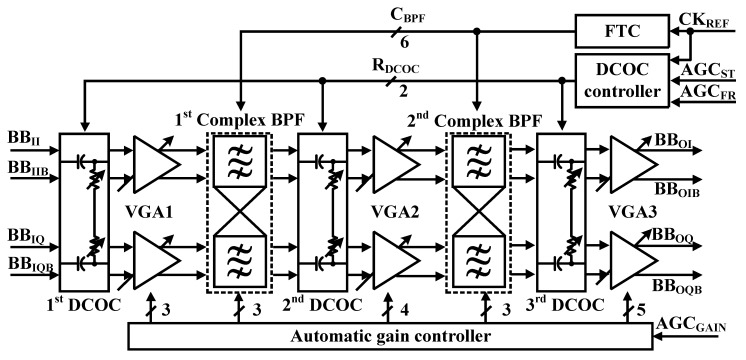
Block diagram of low IF baseband.

**Figure 7 sensors-19-02420-f007:**
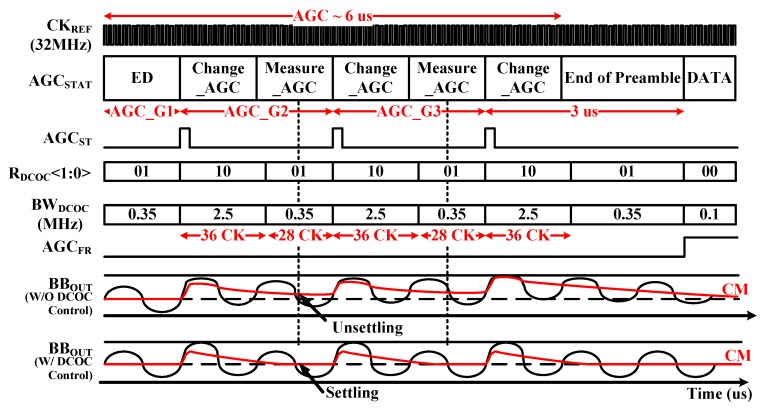
Timing diagram of the DCOC Controller.

**Figure 8 sensors-19-02420-f008:**
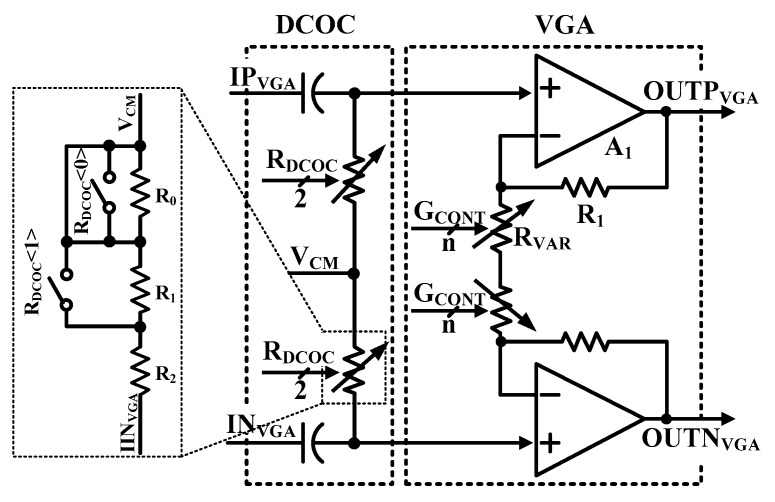
Schematic of variable gain amplifier with DCOC.

**Figure 9 sensors-19-02420-f009:**
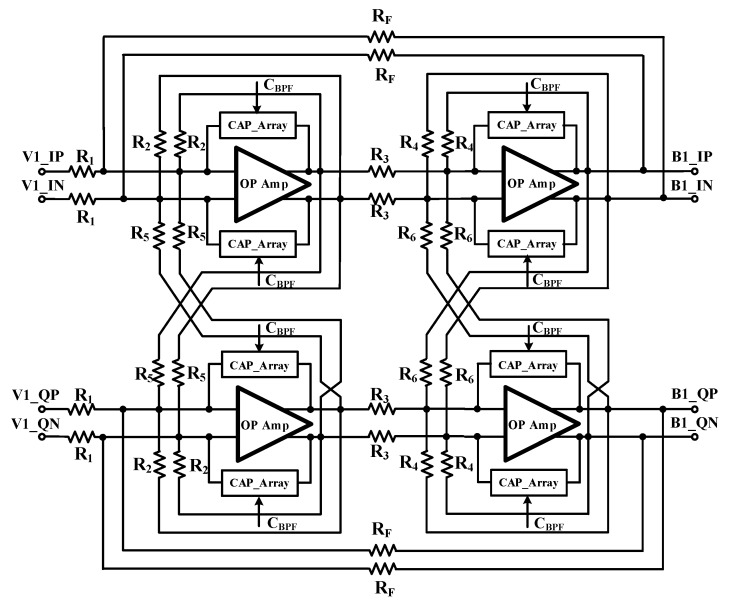
Block diagram of the second order complex BPF.

**Figure 10 sensors-19-02420-f010:**
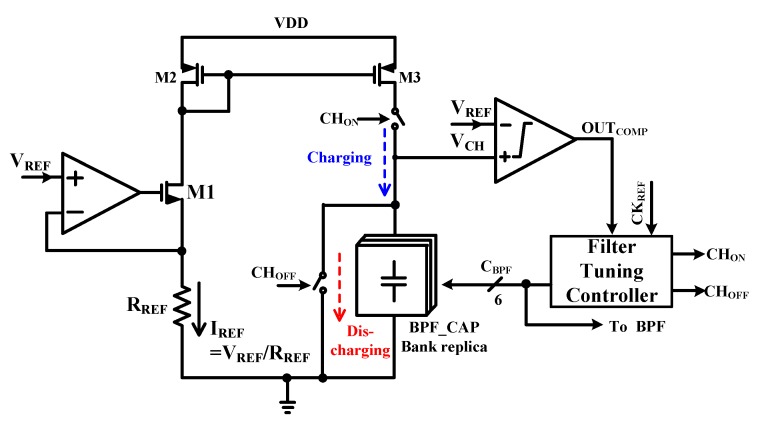
Schematic of the RC filter tuning circuits.

**Figure 11 sensors-19-02420-f011:**
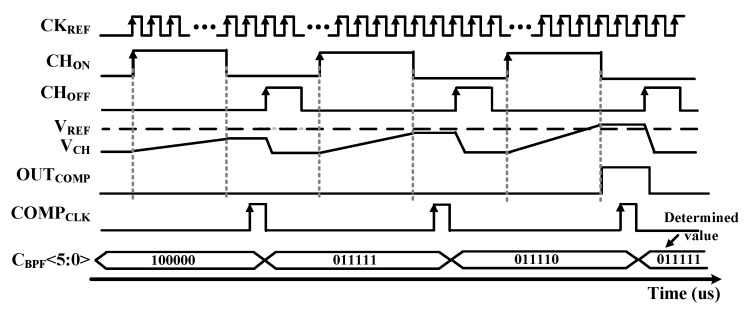
Timing diagram of the RC filter tuning circuit.

**Figure 12 sensors-19-02420-f012:**
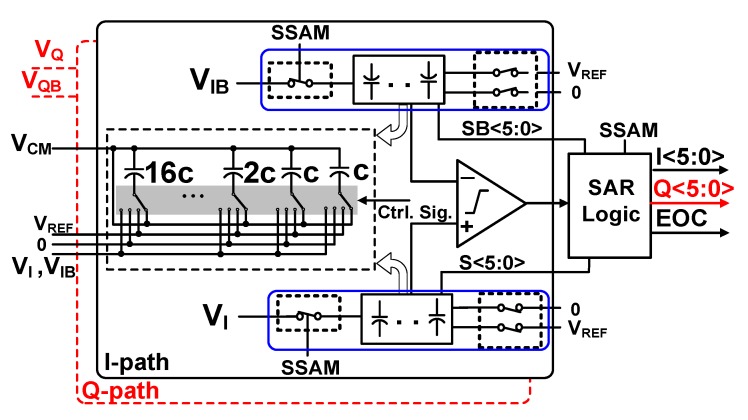
Block diagram of proposed fully differential SAR ADC.

**Figure 13 sensors-19-02420-f013:**
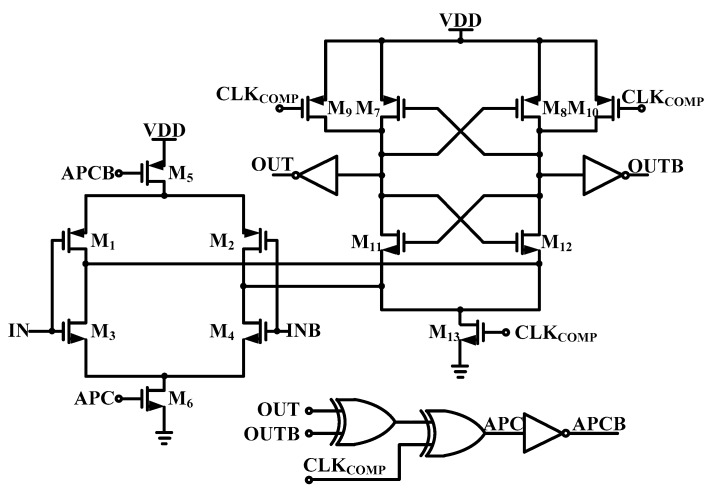
Schematic of the proposed dynamic latched comparator.

**Figure 14 sensors-19-02420-f014:**
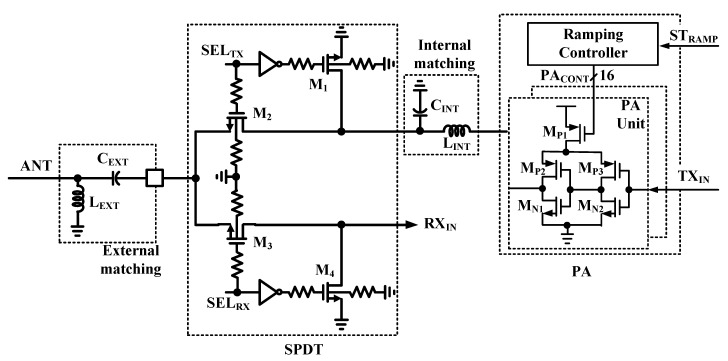
Block diagram of PA and SPDT.

**Figure 15 sensors-19-02420-f015:**
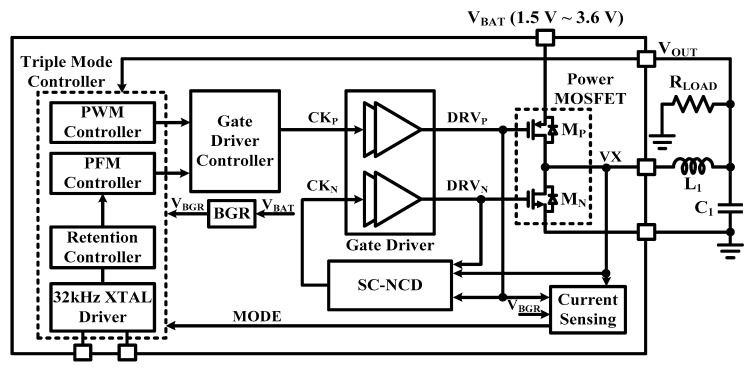
Block diagram of the DC-DC buck converter in triple-mode.

**Figure 16 sensors-19-02420-f016:**
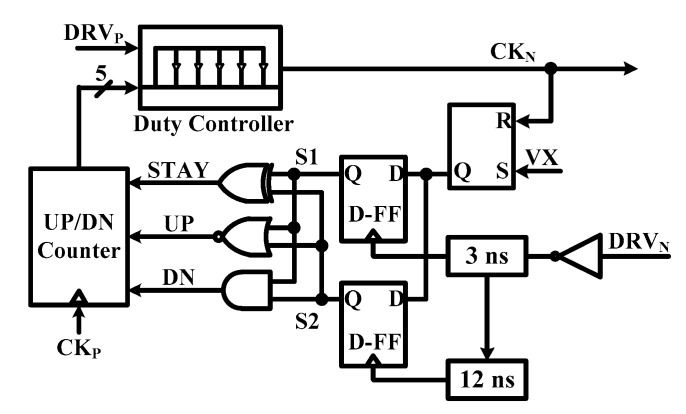
Schematic of proposed SC-NCD.

**Figure 17 sensors-19-02420-f017:**
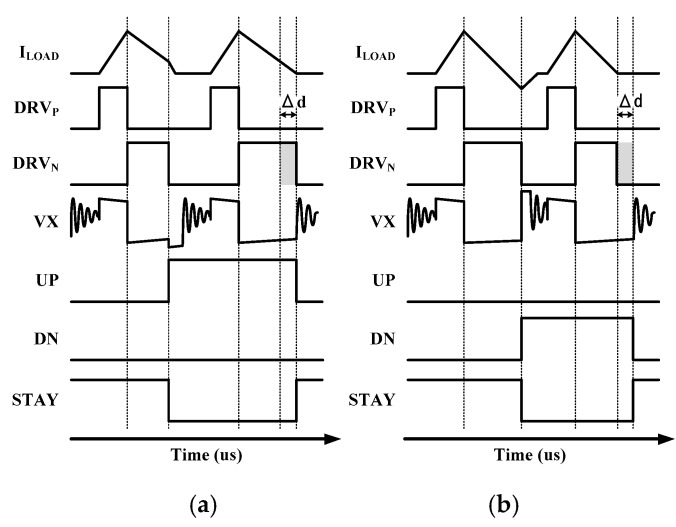
Timing diagram of SC-NCD (**a**) under duty operation, and (**b**) over duty operation of the low side.

**Figure 18 sensors-19-02420-f018:**
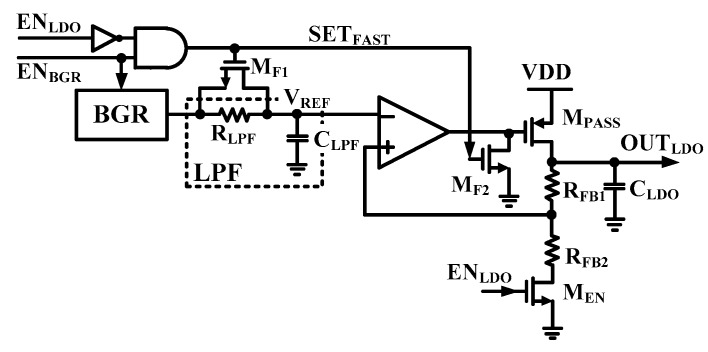
Block diagram of low noise LDO with a fast settling technique.

**Figure 19 sensors-19-02420-f019:**
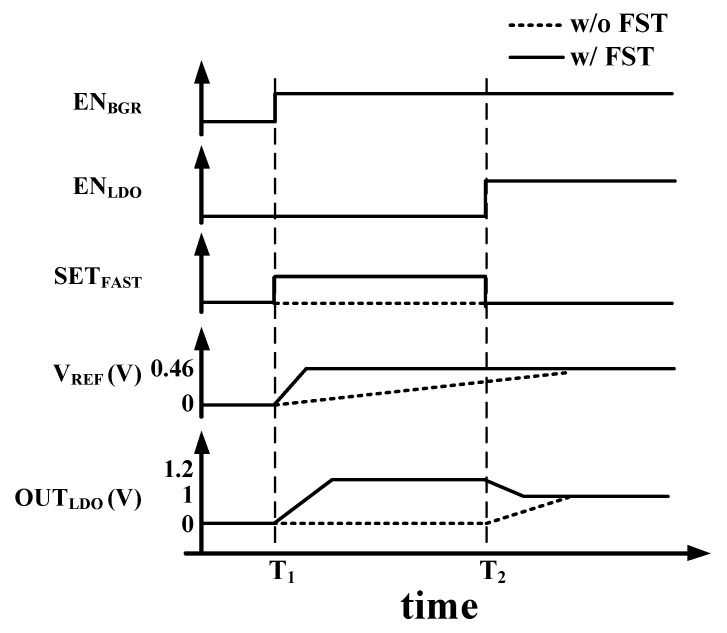
Timing diagram of the fast settling technique.

**Figure 20 sensors-19-02420-f020:**
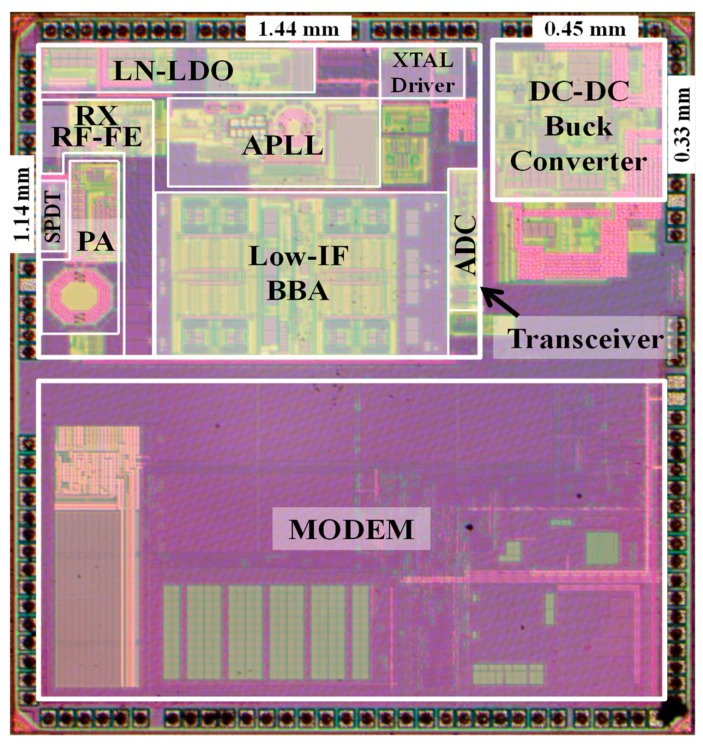
Chip micro-photo of the proposed BLE transceiver.

**Figure 21 sensors-19-02420-f021:**
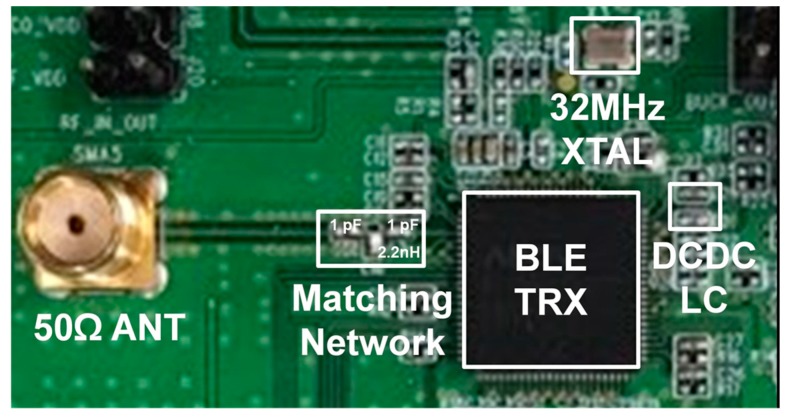
Measurement board of the BLE transceiver.

**Figure 22 sensors-19-02420-f022:**
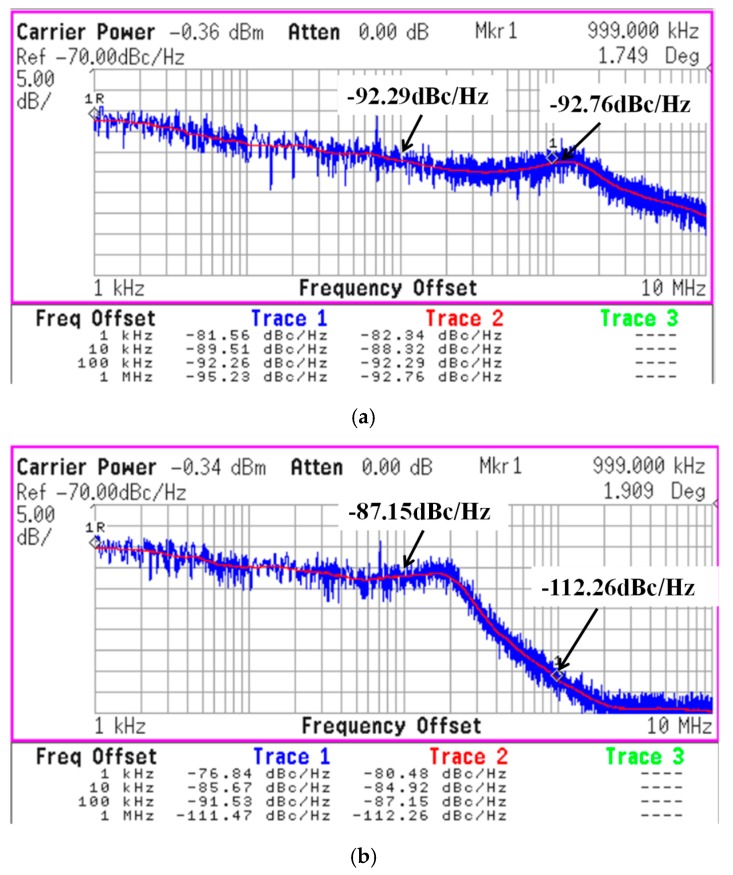
Measured phase noise of the proposed PLL in (**a**) TX mode and (**b**) RX mode.

**Figure 23 sensors-19-02420-f023:**
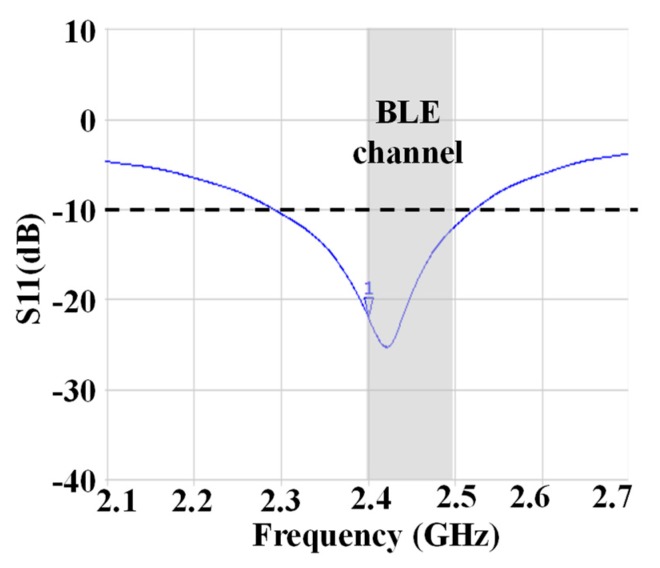
The measurement result of S_11_ characteristics of the proposed transceiver.

**Figure 24 sensors-19-02420-f024:**
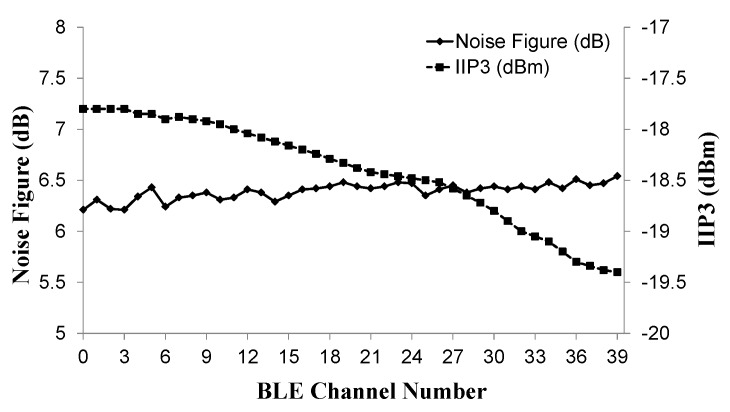
IIP3 and NF measurement results with respect to BLE channels.

**Figure 25 sensors-19-02420-f025:**
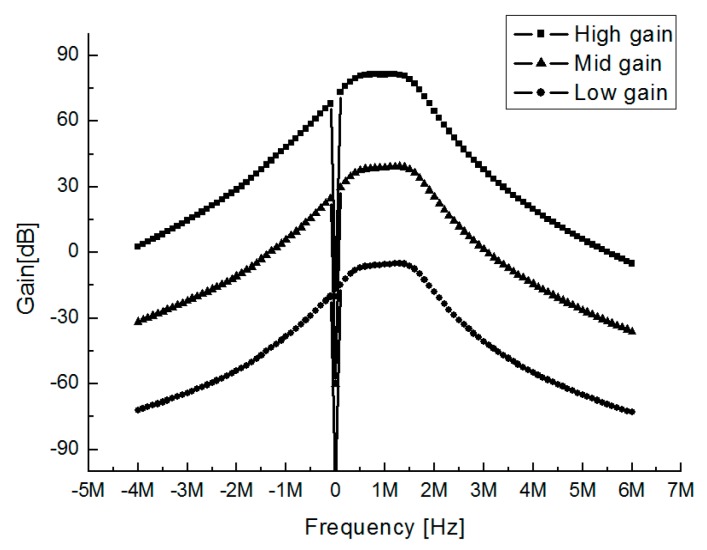
The measured gain and bandwidth of BBA.

**Figure 26 sensors-19-02420-f026:**
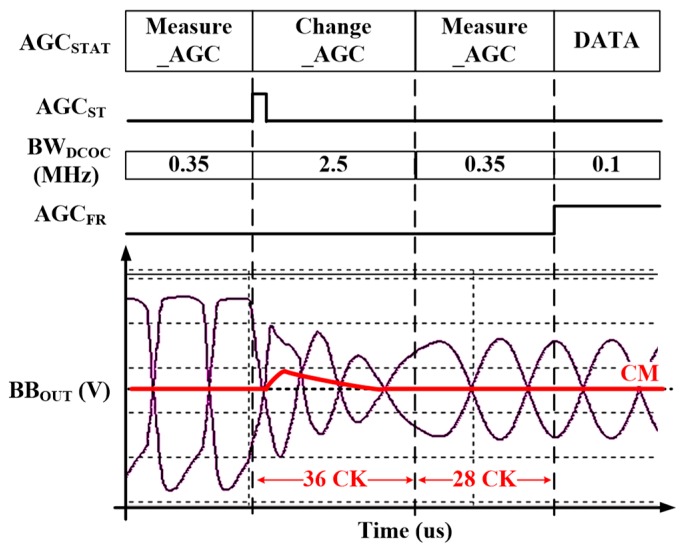
The simulation result of the DC-OC Controller.

**Figure 27 sensors-19-02420-f027:**
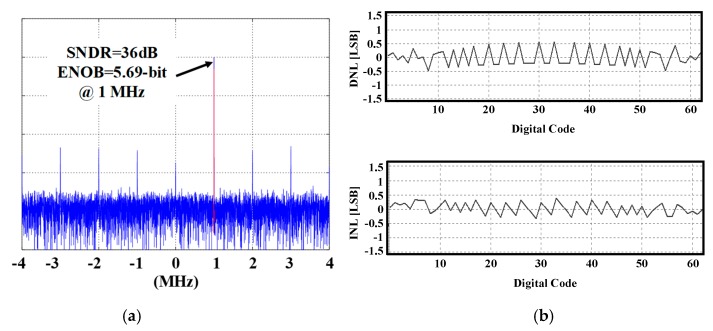
The measured (**a**) ENOB and (**b**) INL and DNL of ADC.

**Figure 28 sensors-19-02420-f028:**
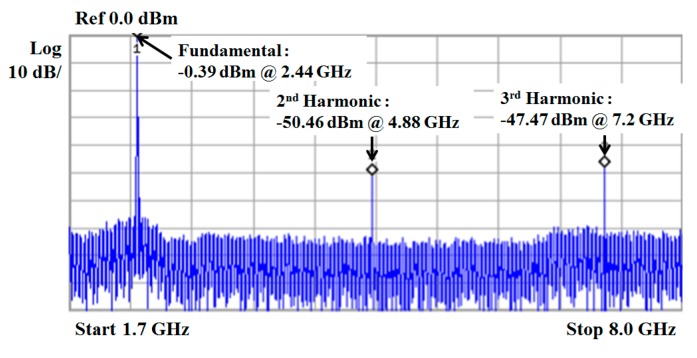
The measured output spectrum of TX.

**Figure 29 sensors-19-02420-f029:**
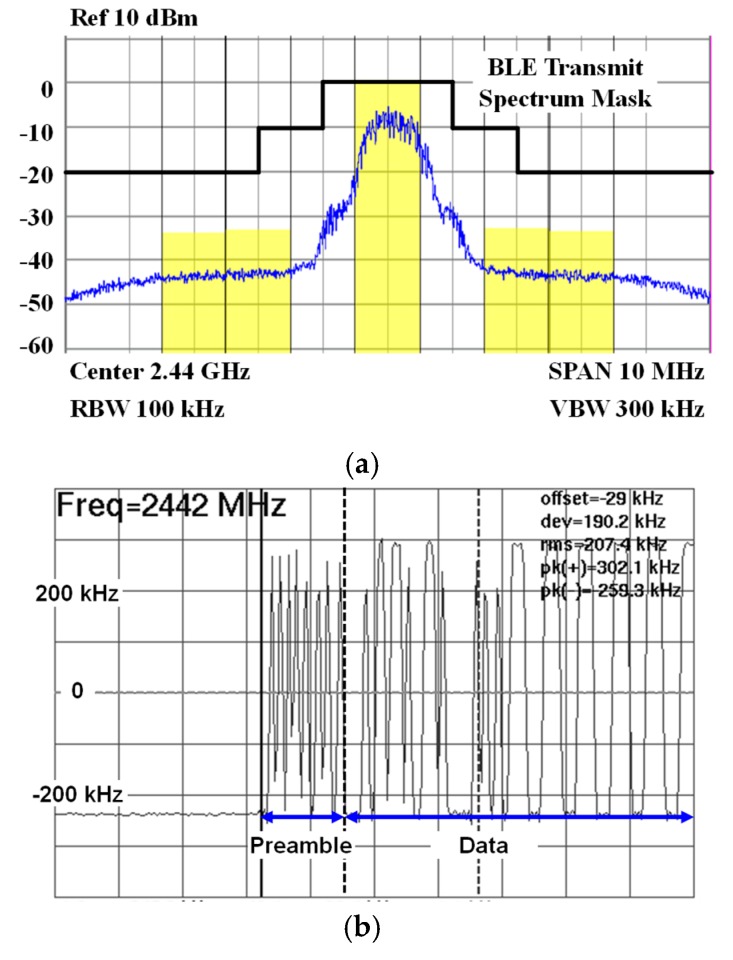
(**a**) Spurious emission mask of TX and (**b**) demodulated TX frequency for the BLE packet.

**Figure 30 sensors-19-02420-f030:**
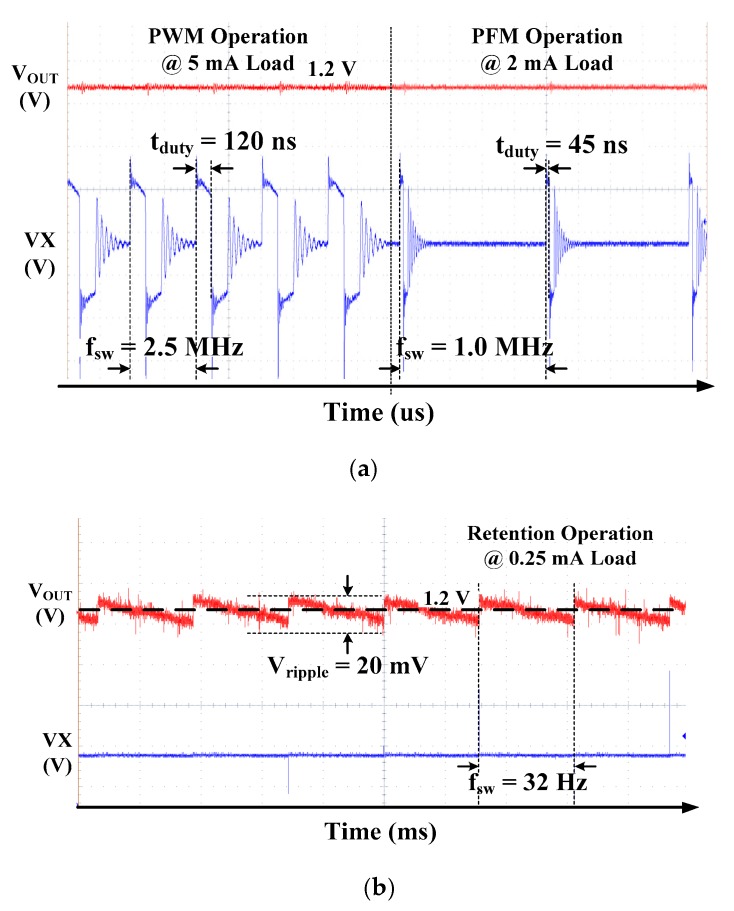
Measured waveforms of the triple-mode DC-DC Converter in (**a**) PWM/PFM mode and (**b**) retention mode.

**Figure 31 sensors-19-02420-f031:**
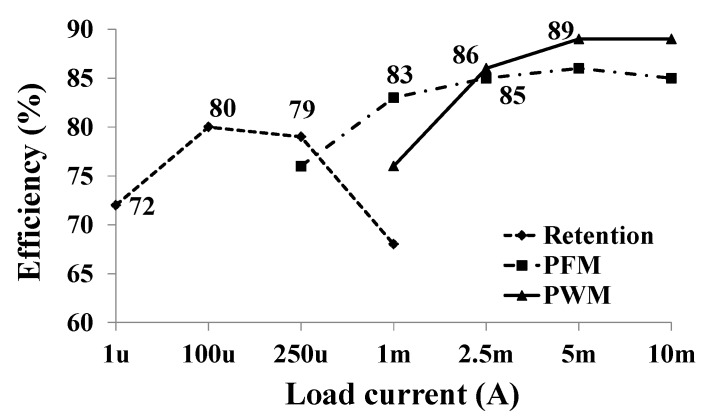
Measured efficiency of the DC-DC buck converter in triple mode.

**Figure 32 sensors-19-02420-f032:**
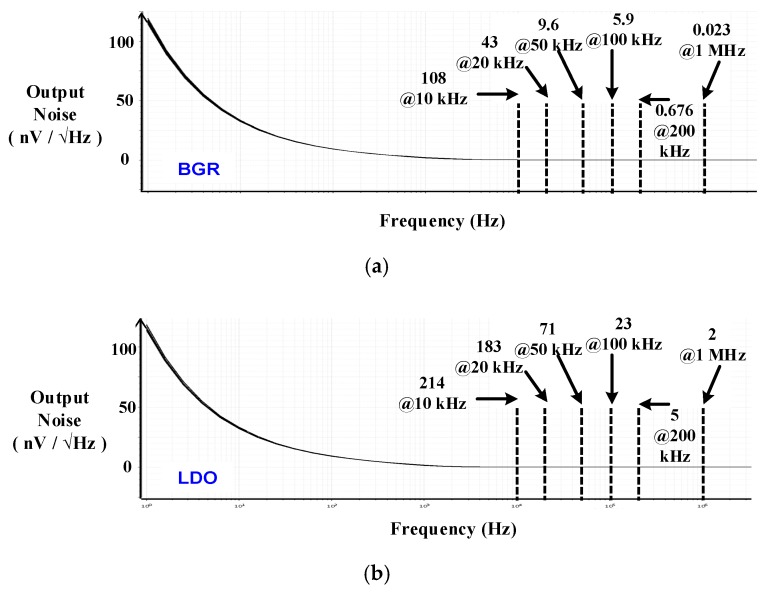
Simulated output noise of (**a**) BGR and (**b**) LDO.

**Figure 33 sensors-19-02420-f033:**
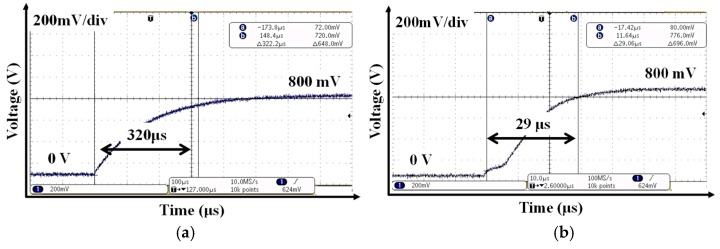
Measured output of the LDO (**a**) with and (**b**) without a fast settling technique.

**Table 1 sensors-19-02420-t001:** Parameter of loop filter in PLL.

Mode	RX	TX
**ω_C_**	200 kHz	1 MHz
**K_VCO_**	50 MHz/V	
**I_CP_**	20 μA	120 μA
**R_1_**	95 kΩ	82 kΩ
**R_2_**	450 kΩ	225 kΩ
**C_1_**	1.4 pF	0.45 pF
**C_2_**	39 pF	15 pF
**C_3_**	250 fF	100 fF

**Table 2 sensors-19-02420-t002:** Performance summary.

Reference	[3]	[4]	[21]	This Work
**Technology**	55 nm CMOS	55 nm CMOS	40 nm CMOS	55 nm CMOS
**Data Rate & Modulation**	1-Mbps GFSK	1-Mbps GFSK	1-Mbps GFSK	1-Mbps GFSK
**Supply** **Voltage (V)**	3.0	1.2	3.0	3.0
**TX Output** **Power (dBm)**	0	8	0	0
**TX power consumption** **(mW)**	10.1 @ 0 dBm	79.6 @ 8 dBm	10.8 @ 0 dBm	6 @ 0 dBm
**RX Architecture**	Low-IF	Low-IF	Zero-IF	Low-IF
**RX Power Consumption (mW)**	11.2	12	6.3	5
**RX Noise** **Figure (dB)**	5.5	N/A	6.5	6.52
**RX Sensitivity (dBm)**	−94.5	−96.8	−94.5	−95
**RX IIP3 (dBm)**	N/A	N/A	N/A	-17.8
**Area (mm^2^)**	2.9	2.2	1.1	1.79

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
