# Peer review of "A Fully Integrated Bluetooth Low-Energy Transceiver with Integrated Single Pole Double Throw and Power Management Unit for IoT Sensors"

_sensors, 2019, doi:10.3390/s19102420_

Round 1
Reviewer 1 Report
This paper presents an implementation of a complete, very low power BLE transceiver. The transceiver was build and extensively tested with very good comparative results.
I could only find a few typos of the text:
- GF for gaussian filter is not defined in the text;
- Line 86 - instead of [8][9][10] please use [8-10];
- Lines 91 and 92 - The phase is difficult to understand, please rewrite;
- Line 107 - please use RX instead of Rx;
- Lines 109 and 110 - It is not clear how Cm can prevent noise from TIA;
- Line 159 - There is a typo in the formula. Frequency is misspelled;
- Lines 232 - 233 - The meaning of the sentence is not clear.
- Line 369 - Use "Microphoto" instead of "micrograph";
- Line 380 - Maybe it is interesting to show the values on the matching network components employed;
- Line 390 - In the caption noise in misspelled;
- Line 396 - It is not clear if the S11 values presented are simulated or measured;
- Line 423 - Meaning if the phrase starting with Spurious is not clear. Please rephrase;
- Line 429 - Fig. 29(b) should be referenced instead of Fig. 30(b);
- Lines 446 - 454 - It is not clear why in Fig. 28 maximum output power is -0.39dBm and in Fig. 29 the maximum power is around -7dBm. I assume the integration over the bandwidth will lead to the 0dBm, but please explain;
- Lines 477 - No clues are given in the text of how these values were obtained.
Just a final remark: no future work is presented. I would really like to see a BER evaluation between two separated devises.
Is is my opinion that the paper and the work are both very good and is should be accepted after the minor correction I suggest.
Author Response
Dear editor and reviewers,
Following revisions were made based on the reviewer’s comments and suggestions.
We highlighted the changed parts in the revised manuscript with blue color.
Reviewers' and Associate Editor's Comments
- Common
1. The reference numbers are changed before and after the revision as follows.
[5] S. Oh et al., “A 3.9 mW Bluetooth Low-Energy Transmitter Using All-Digital PLL-Based Direct FSK Modulation in 55 nm CMOS,” IEEE Trans. Circuits Syst. I, Reg. Papers, 2018 vol. 65, no. 9, pp. 3037–3048.
[12] Keliu Shu and Edgar Sanchez-Sinencio, “CMOS PLL Synthesizers: Analysis and Design,” Springer.
[18] CIRCUIT NOTE, “Powering a Fractional-N Voltage Controlled Oscillator with Low Noise LDO Regulators for Reduced Phase noise,” ANALOG DEVICE, CN-0147.

Reviewer 2 Report
This paper describes a BLE TRX design with solid technical performance. By including SPDT as well as power management unit, the presented chip achieves high integration low cost design. I have a few minor comments.
- Please explain why analog PLL is designed in this work. Since this paper proposes low cost design with minimum external component and low area, the use of the digital PLL would have been considered.
- The analog PLL has two BW settings. The authors need to explain whether there is any issue of settling time issue when the PLL in RX mode with 200kHz BW changes the mode for TX or vice versa.
- ENOB of ADC is less than 6 bit. Please describe more about system design considerations with <6b ENOB.
Author Response
Reviewer Responses:
This paper describes a BLE TRX design with solid technical performance. By including SPDT as well as power management unit, the presented chip achieves high integration low cost design. I have a few minor comments.
Point 1: - Please explain why analog PLL is designed in this work. Since this paper proposes low cost design with minimum external component and low area, the use of the digital PLL would have been considered.
Response 1:
- Thank you very much for your valuable comments
- As the reviewer pointed out, digital PLL can have benefits in terms of the external components and the low area. However, additional calibration digital logics and circuits are required for the ADPLL to meet the PVT variation specification and the frequency deviation specification of the BLE application.
- For the simplicity of implementation, we used the analog PLL instead of the digital PLL.
- We added the following comments to page 2, line 71 of the revised manuscript.
- “Although the digital PLL can have the benefit in terms of the area, analog PLL is implemented in this paper to avoid the complexity of additional calibration logics in digital PLL [5]”
Point 2: The analog PLL has two BW settings. The authors need to explain whether there is any issue of settling time issue when the PLL in RX mode with 200kHz BW changes the mode for TX or vice versa.
Response 2:
- Thank you very much for your valuable comments
- As the reviewer pointed out, PLL lock time between the TX and RX is different because of the bandwidth.
- Since the PLL lock time is approximately 4/(Bandwidth) and the RX bandwidth is 200 kHz, the PLL lock time is about 20 ms when it is changed from the TX to RX. In the BLE specification, TX/RX switching time is 150 ms. As a result, BLE specification can be satisfied with the different bandwidths of TX and RX in this paper.
- We added the following comments to page 4, line 136 of the revised manuscript.
- “As can be seen from the table, the TX and RX bandwidths are different from each other. The PLL lock time is approximately 4/(Bandwidth) which makes the RX PLL locking time about the 20 ms [12]. Since the TX/RX switching time is 150 ms from the BLE specification [2], the RX PLL locking time can satisfy the BLE specification with the certain margin.”
Point 3: ENOB of ADC is less than 6 bit. Please describe more about system design considerations with <6b ENOB.
Response 3:
- Thank you very much for your valuable comments.
- We added the following sentences in page 9, line 271 of the revised manuscript.
- “The resolution of ADC required in modem requires 5-bit, but was designed with 1-bit margin when designing the SAR ADC.”
Thank you very much.
Corresponding Author

Reviewer 3 Report
The paper describes a Bluetooth Low-Energy Transceiver with power management capabilities. The presented results are complete and well presented.
However, the authors fail to present a basic state of the art in which related work is presented. The authors fail to identify any novelty and describe in which implemented characteristics their work presents novelty.
Also the option for different architecture modules should be substantiated.
Author Response
Reviewer Responses:
The paper describes a Bluetooth Low-Energy Transceiver with power management capabilities. The presented results are complete and well presented.
However, the authors fail to present a basic state of the art in which related work is presented. The authors fail to identify any novelty and describe in which implemented characteristics their work presents novelty.
Also, the option for different architecture modules should be substantiated.
Response:
- Thank you very much for your valuable comment.
- We described the novelty of this manuscript as follows and added sentences to the revised manuscript.
1) In the PLL, Automatic Bandwidth Calibration is proposed for the accurate PLL bandwidth setting which can be applied both for TX and RX.
2) In the Baseband Analog (BBA), Filter Tuning Circuit is proposed for the accurate filter characteristics regardless of the PVT variation.
3) In the ADC, Adaptive Power Control is proposed to minimize the current consumption of the comparator.
4) In the TX Power Amplifier, Digital Ramping Control algorithm is proposed to reduce the spurious tones.
5) Along with the ideas we’ve mentioned above, this paper also integrated the Power Management Unit comprising of the DC-DC buck converter and LDOs. The previous papers related to the Bluetooth Low-Energy Transceiver were mostly focused on the PLL or the RF frontend. In this paper, the Self-Calibration Negative Current Detector is proposed in the DC-DC buck converter to enhance the efficiency. Moreover, Fast Settling Technique is proposed in the LDO for the fast settling with the low noise.
Thank you very much.
Corresponding Author
